



# Characteristics and evolution of diurnal foehn events in the Dead Sea valley

Jutta Vüllers[1], Georg J. Mayr[2], Ulrich Corsmeier[1], and Christoph Kottmeier[1]

[1]Institute of Meteorology and Climate Research, Karlsruhe Institute of Technology (KIT), POB 3640, 76021 Karlsruhe, Germany
[2]Department of Atmospheric and Cryospheric Sciences University of Innsbruck, Innrain 52f, 6020 Innsbruck, Austria

*Correspondence to:* J. Vüllers (jutta.vuellers@kit.edu)

**Abstract.** This paper investigates frequently occurring foehn in the Dead Sea valley. For the first time, sophisticated, high-resolution measurements were performed to investigate the horizontal and vertical flow field. In up to 72 % of the days in summer, foehn was observed at the eastern slope of the Judean Mountains around sunset. Furthermore, the results also revealed that in approximately 10 % of the cases the foehn detached from the slope and only effected elevated layers of the valley

atmosphere. Lidar measurements showed that there are two main types of foehn. Type I has a duration of approximately 2-3 h and mean maximum velocities of around $5\,\mathrm{m\,s^{-1}}$ and does not propagate far into the valley, whereas type II affects the whole valley, as it propagates across the valley to the eastern side. Type II reaches mean maximum wind velocities of $11\,\mathrm{m\,s^{-1}}$ and has a duration of about 4-5 h. A case study of a type II foehn shows that foehn is initiated by the horizontal temperature gradient across the mountain range. In the investigated case this was caused by an amplified heating and delayed cooling of the valley

boundary layer in the afternoon, compared to the upstream boundary layer over the mountain ridge. The foehn was further intensified by the advection of cool maritime air masses upstream over the coastal plains leading to a transition of subcritical to supercritical flow conditions downstream and the formation of a hydraulic jump and rotor beneath. These foehn events are of particular importance for the local climatic conditions, as they modify the temperature and humidity fields in the valley and, furthermore, they are important because they enhance evaporation from the Dead Sea and influence the aerosol distribution in

the valley.

## 1  Introduction

In mountainous terrain the atmospheric boundary layer, and thus the living conditions in these regions, are governed by processes of different scales. Under fair weather conditions, the atmospheric boundary layer (ABL) in a valley is often decoupled from the large-scale flow by a strong temperature inversion (Whiteman, 2000). In this case mainly local convection and ther-

mally driven wind systems, which are caused by differential heating of adjacent air masses, such as slope and valley winds, determine the valley ABL (e.g. Atkinson, 1981; Zardi and Whiteman, 2013). They influence the diurnal temperature and humidity cycle (e.g. Alpert et al., 1997; Bischoff-Gauß et al., 2008) and also determine aerosol dispersion and, thus, air quality (e.g. Kalthoff et al., 2000; Corsmeier et al., 2005; Fast et al., 2006). Mesoscale processes, such as topographic and advective venting accompany the thermally driven flows under fair weather conditions (e.g. Fast and Zhong, 1998; Adler and Kalthoff,



2014). When the large scale flow is not negligible it also impacts the valley ABL. The interaction takes place via turbulent transport or dynamically driven flow phenomena, which occur when the large scale flow is effected by the orography. Gravity waves, wave breaking, downslope windstorms, hydraulic jumps and rotors can occur on the lee side of the mountains. The intensity and extent of the developing phenomena depends on the shape of the mountains, the stratification of the atmosphere,

the strength of the valley ABL inversion, the wind speed, and the direction of the large-scale flow (Whiteman, 2000). Stratified flow theory as well as hydraulic flow theory were both used successfully to explain the aforementioned phenomena (e.g. Schär and Smith, 1993; Corsmeier et al., 2005; Durran, 2003; Jackson et al., 2013). Field campaigns were performed to gather observations of these phenomena along large mountain barriers (Alps, Pyrenees) during field campaigns such as ALPEX (Davies and Pichler, 1990), PYREX (Bougeault et al., 1990) or MAP (Bougeault et al., 2001) and also for individual valleys, e.g.

T-REX (Grubišic et al., 2008), MATERHORN (Fernando et al., 2015) or craters like in METCRAX (Whiteman et al., 2008; Lehner et al., 2016). This paper contributes to this research by investigating a frequently occurring mesoscale flow phenomena which influences weather and climate in the Dead Sea (DS) valley.

The Dead Sea with a water level of currently -430 m above mean sea level (amsl) forms the lowest part of the Jordan Rift Valley, which is an over 700 km long north-south oriented depression zone extending from the northern Israeli border to the

Gulf of Aqaba. The complex and steep orography, together with the land surface heterogeneity in the valley, introduced by the lake, results in a strong local forcing and triggers pronounced thermally driven wind systems, such as a lake breeze, slope, and valley winds (Kottmeier et al., 2016). Additionally, regional forcing influences the atmospheric processes in the valley, resulting in a distinct diurnal wind pattern. In particular, strong westerly downslope winds are observed frequently in the evening. Ashbel and Brooks (1939) first described the westerly winds in the northern part of the DS as the Mediterranean Sea Breeze

(MSB) entering the valley, but not with the typical characteristics of a sea breeze at the coast, a steady, cool and moist air flow, they rather described it as a very dry, hot and gusty wind. Following Ashbel and Brooks (1939) various observational near surface studies (e.g. Bitan, 1974, 1976; Lensky and Dayan, 2012; Naor et al., 2017) and also numerical studies (e.g. Doron and Neumann, 1978; Alpert et al., 1982; Segal et al., 1983, 1985) have been carried out to study the penetration of the MSB into the Jordan rift valley. Studies showed that the downward penetration of the MSB results from the temperature difference

between the cooler maritime air mass and the warmer valley air mass. A density-driven flow, which accelerates and warms while descending into the valley (Alpert et al., 1990), i.e. a foehn wind, when following the definition of the WMO (WMO, 1992). At the DS they occur most frequently in summer and enter the Jordan rift valley first in the North around Lake Kinneret and approximately at 18:00 LT (LT=UTC+2) at the DS (Bitan, 1974, 1976; Segal et al., 1985; Naor et al., 2017). These foehn events have a large impact on the atmospheric conditions in the DS valley. Mean hourly wind velocities of $5\,\mathrm{m\,s^{-1}}$

in the south and up to $12\,\mathrm{m\,s^{-1}}$ in the north were recorded during these events (Bitan, 1974; Weiss et al., 1988; Hecht and Gertman, 2003). Furthermore, through the adiabatic heating of the air mass during the descent into the valley, the foehn warms the valley (Ashbel and Brooks, 1939; Alpert et al., 1997; Shafir and Alpert, 2011). It also influences humidity. Some studies suggest an increase in humidity (Naor et al., 2017), whereas others have suggested a decrease (Ashbel and Brooks, 1939). The foehn events influence the lake evaporation, as evaporation is driven by wind velocity and vapour pressure deficit as shown by

Metzger et al. (2018). The diurnal maximum of evaporation is reached, untypically, shortly after sunset when the foehn sets in



(Metzger et al., 2018). Finally, the foehn events can also cause an air mass exchange and remove the aerosol particles and the often occurring haze layer from the valley (Levin et al., 2005; Holla et al., 2015).

Even though these foehn events have apparently such a big influence on the atmospheric conditions at the DS, a detailed analysis of their three-dimensional structure and their characteristics, such as height, duration and intensity, and further insights regarding their evolution are missing. Hence, the following questions are addressed in this study: (i) Is a differentiation between radiation and density-driven downslope flows possible? (ii) What are the typical characteristics of the foehn events? (iii) What are the key mechanisms during the foehn evolution?

For the analysis the first sophisticated, high-resolution lidar measurements of foehn events in the DS valley, along with long-term near-surface observations were used. The measurements were performed in the framework of the interdisciplinary virtual institute DEad SEa Research VEnue (DESERVE) (Kottmeier et al., 2016). In the following section, Sec. 2, a short geographical overview, information on the measurement sites and the instrumentation as well as the applied methods are presented. Sec. 3.1 presents results of an objective occurrence frequency analysis, characteristics of the foehn events are shown in Sec. 3.2 and a detailed case study of a strong foehn event and the processes leading to it are presented in Sec. 3.3. Sec. 4 provides a summary, including a conceptual model of the processes, and conclusions.

## 2 Methodology

### 2.1 Study area

The DS is the lowest reachable place on earth with a current water level of -430 m amsl. The valley is North-South oriented, with the Judean Mountains to the west, with a mean ridge height of about 895 m amsl and the Moab Mountains to the east, which reach up to 1200 m amsl (Fig. 1 a). The cross section in Fig. 1 b illustrates the steep orography at both sides of the DS extending over 1600 m in the vertical. The DS is about 100 km long and 15 to 17 km wide. The distance to the Mediterranean Sea is 80 km in the north and increases up to 120 km towards the south (Fig. 1 a, b).

### 2.2 Experimental Setup

A long-term meteorological observation network is operated around the Dead Sea since 2014 (Fig. 1). It was complemented by special observation periods in August 2014 and November 2014, where the mobile observatory 'KITcube', an assembly of ground-based in-situ and remote sensing instruments for probing the atmosphere (Kalthoff et al., 2013), was deployed in the DS valley, near Masada (Fig. 1 a, c). Additional radiosoundings were performed at the eastern shore of the DS in Ghor Haditha, Jordan (Fig. 1 a). Further details about the measurement network around the DS and the KITcube instruments can be found in Kottmeier et al. (2016) and Kalthoff et al. (2013).





## 2.3 Instrumentation and data

The long-term measurements were performed using meteorological towers which were equipped with sensors measuring air temperature, humidity, shortwave and longwave radiation components of the upper and lower half space at 2 m above ground level (agl), air pressure, and precipitation at 1 m agl, with a temporal resolution of 10 min. Supplementary measurements with a temporal resolution of 20 Hz were performed with a sonic anemometer at 10 m agl at the meteorological stations (Fig.1 a, orange stations) and with an integrated gas analyser and sonic anemometer (IRGASON) at 6 m agl at the energy balance stations (Fig.1 a, red stations), providing all three wind components, sonic temperature, and water vapour. Furthermore, long-term observations of wind, temperature, and pressure provided at Jerusalem by the Israel Meteorological Service (IMS) and the operational radiosoundings from the IMS at Bet Dagan launched everyday at 00:00 and 12:00 UTC were used.

The following subset of KITcube instruments was used in this study. Aerosol backscatter and radial velocity from a two-axis scanning pulsed 2 μm wind lidar from Lockheed Martin called 'WindTracer', with a peak power of 4.5 kW and a pulse repetition frequency of 500 Hz (Träumner, 2012). The effective pulse length and hence the minimum spatial resolution is 56 m. The scan pattern of the instrument included range-height-indicator (RHI) scans at azimuth angles of 15, 62, 172, and 299° with an elevation ranging from 0 to 180°, and plan-position-indicator (PPI) scans of 360° azimuth for elevation angles of 0.2, 5, 15, and 75°. The minimum detection range of the instrument is 400 m. A quality threshold of -7 dB was used for the Signal-to-Noise-Ratio (SNR). To close the gap between the surface and the minimum detection range of the Windtracer, a second wind lidar from Leosphere was used. The 'Windcube' has a resolution of 20 m and a detection range from 40 m to 600 m. It operates with a wavelength of 1.54 μm and works with a 4-point stop-and-stare mode at a fixed elevation angle of 75.2° and an integration time of 7 s. From both lidar instruments, the vertical profiles of the horizontal wind were calculated using the VAD algorithm after Browning and Wexler (1968). Radiosondes were simultaneously launched during intensive observation periods on both sides of the DS every 2 h always from Friday 14:00 LT until Sunday 06:00 LT, due to airspace restrictions, providing temperature, humidity, wind velocity and wind direction profiles.

## 2.4 Layer detection algorithm

The characteristics of the foehn events were derived from lidar range-height-indicator (RHI) scans along the main wind direction of the foehn events (black dashed line in Fig. 2 a). By introducing a layer detection algorithm the characteristics could be derived objectively. The algorithm uses profiles of the horizontal component of the radial velocity (Fig. 2 b) and the backscatter signal (Fig. 2 c) averaged between 2 and 3 km away from the lidar. The profiles show that within the foehn layer the flow has a jet-like structure with strong wind velocity and backscatter gradients (Fig. 2 d). The strongest wind velocity and backscatter gradient above the wind velocity maximum and the backscatter minimum, respectively, were used within the algorithm to determine the foehn layer height (red dashed line in Fig. 2 d). Furthermore, the algorithm detected the mean and maximum wind speed, and the height of the wind speed maximum from each RHI scan. These results were then used to derive averaged characteristics for each event.





## 2.5   Reduced-gravity shallow-water theory

Mountains can have a large influence on the wind field. An air mass can either be blocked and forced to flow around the mountain, it can flow over the mountain or it can pass through gaps in the mountain rainge. The ratio between the energy needed to overcome negative buoyancy and the horizontal inertia of the air mass (Jackson et al., 2013) is called the non-dimensional mountain height

$$\hat{H} = \frac{Nh}{U}, \tag{1}$$

with the crest height, $h$, the horizontal wind speed upstream, $U$, and the Brunt-Vaisala frequency, $N$. $\hat{H}$ is normally calculated for a layer, and thus the values represent mean values of this layer. For low wind speeds or strongly stable stratification the air mass is blocked by the mountain indicated by $\hat{H} > 1$, but for $\hat{H} \leq 1$ the flow has sufficient kinetic energy to flow over the mountain. An important theory, with which such dynamically driven flows over mountains can be described, is the reduced-gravity shallow-water (RGSW) theory as described by Pettre (1982). It describes a two-layer system separated by a strong temperature inversion. To explain the characteristics of the flow the Froude number,

$$Fr = \frac{u}{\sqrt{g'\eta}} \tag{2}$$

where $u$ is the mean wind velocity of the lower layer, $\eta$ the inversion height, and $g' = g \cdot \Delta\Theta/\Theta$, the reduced gravitational acceleration, can be used. $\Delta\Theta$ is hereby, the inversion strength at the top of the lower layer and $\Theta$ the mean potential temperature of the lower layer. The flow is subcritical if $Fr < 1$, $Fr > 1$ means the flow is supercritical and $Fr = 1$ is called the critical state. Three situations can occur when a flow passes an obstacle. Firstly, the layer is subcritical everywhere. Then the layer height has to decrease when the terrain rises and it has the shallowest depth at the crest. Secondly, the flow is supercritical everywhere, then the layer height and the terrain height rise simultaneously. Thirdly, the layer changes from a subcritical state upstream to supercritical state downstream. This means that the layer has to be thick and slow upstream and fast and shallow downstream. Further downstream the layer eventually thickens and slows down in a hydraulic jump. This causes strong turbulence as the momentum is conserved but kinetic energy dissipated (Jackson et al., 2013). This is one of the oldest conceptual models and was first proposed by Long (1954), who suggested that there is a fundamental similarity between downslope wind storms and hydraulic jumps. For the transition of the state from subcritical to supercritical, the flowing layer has to reach its critical state $Fr = 1$ at crest height.

## 3   Results

### 3.1   Occurrence frequency

To analyse how often the occurring westerly downslope winds are actually density-driven the occurrence frequency of foehn was calculated using the automatic and probabilistic statistical mixture model of Plavcan et al. (2014) with data from the long-term observations between 2014 and 2016.





This model enables a probabilistic distinction between the density-driven events and other downslope winds, such as nocturnal radiatively-driven downslope flows. The model is based on the assumption that the probability density function of these two wind regimes are statistically distinguishable in one or more characteristic variables. The model fits parametric distributions for the two wind regimes whose mixture results in the observed probability distribution of all downslope wind cases. It yields

the probabilities that measurements at a particular time are density-driven or radiatively driven downslope flows (further details see Plavcan et al. (2014)). The wind regimes are identified using the temperature difference between the crest (Jerusalem, 810 m amsl) and a downstream station as well as the wind speed at the downstream station. The only parameter which has to be set prior to the fully automatic classification is the wind direction sector indicating "downslope". Probabilities that a downslope flow is density-driven were calculated for two stations, one at the slope (Masada, -7 m amsl, downslope sector 200-315°) and

one in the valley (Ein Gedi Beach, -427 m amsl, downslope sector 220-320 degrees) (Fig. 1 a). Figure 3 a, b show the likelihood of foehn for the temperature difference and the wind velocity in the valley. The two wind regimes can easily be distinguished. During nocturnal downslope flows radiative cooling leads to a stable stratification of the valley ABL, resulting in high positive temperature differences of up to 13 K in the valley and 7 K at the slope, and wind velocities between 1-3 m s$^{-1}$. This coincides with literature values were maximum wind velocities are around 1-5 m s$^{-1}$ for radiative driven downslope flows (Whiteman,

2000; Zardi and Whiteman, 2013). During foehn events, the air mass at crest height descends into the valley, leading to a smaller temperature gradient between -4 to 2 K at the slopes and -3 to 2 K in the valley. Wind velocities accelerate down the slope and can reach up to 16 m s$^{-1}$. The climatological distribution of foehn (Fig. 3 c, d) shows that it occurs most often in the afternoon in summer. They do not occur before 15:00 LT at the slopes and 16:00 LT in the valley. The occurrence frequency is highest directly after sunset with 72 % at the slopes and still 64 % in the valley. In the valley the foehn generally ceases around

midnight whereas at the slope foehn is still observed in up to 16% of the nights. In summer it ceases shortly after sunrise, whereas in winter foehn might continue after sunrise. Thus, on a diurnal as well as on an intra-annual basis foehn is observed more frequently at the slope than in the valley. There are two mechanisms which could be responsible for these differences. Firstly, the foehn detaches from the slope, as it reaches the level of neutral buoyancy in the valley and continues flowing in an elevated layer above the valley floor (Mayr and Armi, 2010). Secondly, strong along-valley flows dominate and disturb the

penetration of foehn into the valley. However, with the long-term near surface observations alone a further investigation of these differences is not possible.

### 3.2   Characteristics of foehn in the Dead Sea valley

To get detailed information about the three-dimensional characteristics of the foehn events, lidar data from the special observation period in August 2014 were analysed. Fig. 4 shows the occurrence frequency of foehn for August 2014 as detected

by the probabilistic statistical mixture model. On 26 days foehn was detected at the slope (blue dots), whereas in the valley only 22 events (red dots) were detected. For 13 events data from the scanning lidar were available. For two events (19 and 27 August), which were detected by the probabilistic model, the scanning lidar was not working during the foehn event, however, the windcube, covering the lowest 40 - 600 m agl, was working and showed a strong westerly flow at the measurement location, confirming the foehn occurrence. The 13 measured events give us the opportunity to derive further characteristics of the foehn




and additionally check the aforementioned hypothesis of elevated foehn layers.

For 10 of the 13 events the characteristics could be derived using the algorithm. Manual inspection of the data for the three other events (17, 24 and 29 August) shows that on these three days the algorithm did not detect the layer characteristics because the foehn was only observed in an elevated layer above the ground at the KITcube measurement site. Only for 29 August radiosondes were available in the afternoon showing a temperature inversion at -220 m amsl and a westerly flow just above this inversion. This supports our hypothesis that the foehn reached the level of neutral buoyancy at an elevated layer above the valley floor (not shown).

The characteristics of the other 10 events, derived with the described algorithm are shown in Table 1. The results of the analysis show considerable differences between the events, regarding their duration, strength and vertical extent. They could be grouped into two main categories. The weak, shallow and short events (i), and the longer-lasting, strong events with a large vertical extent (ii). Two events did not fit in one of the groups as they had a very long duration of over 11 h and varying layer characteristics over the course of the event, therefore called 'mixed events'. An example for the two main types is shown in Fig. 5.

### 3.2.1 Weak, shallow and short events

The first type of foehn is a short and weak foehn wind, which only partly penetrates into the valley. The example from the 22 August 2014 lasted about 2 h from 18:00 to 20:00 LT with maximum near surface wind velocities of about $5\,\mathrm{m\,s^{-1}}$ (Fig. 5 a). Before the west wind reached the valley, an elevated west wind maximum just above crest height (1500 m agl) was observed at 16:00 LT, which then penetrated into the valley over a time of about 2 h until it reached the valley floor at around 18:30 LT (Fig. 5 b). To evaluate how far the foehn penetrated into the valley Hovmoeller diagrams were derived from lidar RHI scans along the main flow axis of the foehn using the lowest elevation angle above the orography. The Hovmoeller plot in Fig. 5 c shows how the foehn penetrated down the slopes, starting at around 16:00 LT. An opposing upslope wind was still present during this time slowing down the foehn. At around 18:00 LT the upslope wind ceased and the foehn penetrated down into the valley. For a time window of about 1 h the foehn reached approximately 6 km into the valley and was also observable at the KITcube location, which was 3.5 km south-east of the slope (Fig. 5 c). The mean wind speed of the foehn was $4.1\,\mathrm{m\,s^{-1}}$ and the mean layer height 529 m (Table 1). The mean maximum wind velocity was detected at around 300 m agl with a mean wind velocity of $5.6\,\mathrm{m\,s^{-1}}$.

The characteristics of the other 3 events of this type are summarised in Table 1. They all show very similar characteristics. A rather short duration of two to three hours, mean wind velocities of 3 to $4\,\mathrm{m\,s^{-1}}$, except of one event with $6.2\,\mathrm{m\,s^{-1}}$, a mean layer height of 400 to 600 m and the mean maximum velocity in a height of 200 to 400 m. Two of the four events reached the valley floor, and penetrated approximately 6 km into the valley, whereas the other two foehn events were only observed close to the slope and in an elevated layer above the KITcube location (Table 1).



### 3.2.2 Longer-lasting and strong events

The second example is the 16 August 2014 representing a very strong event with wind velocities exceeding $10 \, \mathrm{m \, s^{-1}}$ (Fig. 5 d). Similar to the first case, an elevated west wind maximum around crest height penetrated into the valley, but it already started at 14:00 LT and affected a deeper layer of about 1500 m (Fig. 5 e). It was observed in the valley at 16:30 LT. Another difference compared to the first type is the strong wind velocity increase while the foehn descended into the valley (Fig. 5 d). At crest height velocities of about $6 \, \mathrm{m \, s^{-1}}$ increased to $9 \, \mathrm{m \, s^{-1}}$ in the valley and the layer height decreased over the course of the event from 1500 m to only 350 m. This strong foehn also penetrated further into the valley. It was observed at the KITcube location for nearly 4 h (Fig. 5 f). The Hovmoeller plot shows that the foehn penetrated at least 13.5 km into the valley, with wind velocities up to $10 \, \mathrm{m \, s^{-1}}$. It could not be determined how far it actually reached into the valley, as the lidar measurements were limited to a radius of 10 km (Fig. 5 f). However, radiosondes launched at the eastern shore of the DS, indicate that the foehn reached towards the other side. Soundings at 17:00 and 19:00 LT also showed north-westerly winds advecting a drier air mass towards the eastern shore, although wind velocities were weaker than on the western side.

The general characteristics of the other strong events were similar to the described one. The overall duration of the strong events varied between 4:00 and 5:30 h, the mean wind velocity of the foehn was 5 to $7 \, \mathrm{m \, s^{-1}}$, and mean maximum wind velocity was 7.5 to $11 \, \mathrm{m \, s^{-1}}$ at a height of about 400 to 570 m. The penetration distance of the four events all exceeded the measurement range of the lidar, meaning that they penetrated at least 13.5 km into the valley. For the 15 and 16 August radiosondes were launched at the eastern shore, confirming that the foehn reached the other side of the valley.

### 3.2.3 Mixed events

The two mixed events had a duration of over 11 h. The other characteristics were similar to the weak events, with mean wind velocities of 4 and $5.4 \, \mathrm{m \, s^{-1}}$, the mean layer height was 251 and 387 m and the mean maximum wind velocity was 6.1 and $8.3 \, \mathrm{m \, s^{-1}}$, respectively. However, the Hovmoeller plot shows, that in particular on the 28 August the event can be divided into two phases. First, a strong downslope wind with velocities of $10 \, \mathrm{m \, s^{-1}}$ persisted for about 4 h between 18:00 and 22:00 LT and then transformed into a shallow layer with wind speed around $5 \, \mathrm{m \, s^{-1}}$ (Fig. 6). For the first 4 h, the foehn penetrated over 13 km into the valley, but in the second phase it merely reached the lidar location. The observations suggest that on 28 August a strong downslope wind event with strong wind speeds transformed into a shallow downslope flow lasting for the rest of the night.

### 3.3 Process understanding of foehn events affecting the whole valley

The foehn events which penetrate far into the valley have obviously the strongest effects on the valley atmosphere. Through the deeper layer and the further penetration into the valley, they cause an air mass exchange, change temperature, humidity and aerosol concentration as well as increase evaporation from the lake surface. The processes which cause such strong events can not be determined by the results presented so far. Therefore, a case study was selected and a detailed analysis of the atmospheric conditions was performed to reveal the relevant processes leading to such strong events As a case study the event of



the 16 August 2014 was selected.

On 16 August a typical summer synoptic system, a shallow Persian Trough, extended from the Persian Gulf over Iraq and Syria bending north-west towards the Mediterranean Sea and Greece (Fig. 7). This resulted in westerly flow in the lowest 1000 m amsl downstream of the trough axis. Offshore over the Mediterranean Sea the near surface winds had an intensity of about 4 m s$^{-1}$ in the morning. The intensity of the trough strengthened in the course of the day and the near-surface westerly flow over the Mediterranean Sea increased to over 7 m s$^{-1}$ in the afternoon.

The evolution of the ABL can be divided into three stages:

### 3.3.1 Stage I: ABL evolution prior to the foehn (7:00-15:00 LT)

In the morning a strong temperature inversion marked the height of the vally ABL at 900 m amsl (Fig. 8 a). In the valley, two local thermally driven wind systems developed. Upslope winds set in around 7:00 LT as indicated by the measurements at the slope (Masada). The wind direction changed from west to east, and the vertical wind velocity changed sign from -0.4 to 0.4 m s$^{-1}$ (Fig. 9). The easterly DS lake breeze reached the KITcube location at 09:00 LT. Wind direction turned east, the 2 m wind velocity increased to 2 m s$^{-1}$, and the diurnal temperature increase over land was attenuated by the advection of the cooler air from the water until 16:30 LT (Fig. 9). The onshore flow of the lake breeze reached up to 350 m amsl with a mean wind speed of 2.4 m s$^{-1}$ and the return flow was observable between 500 and 900 m amsl in the radiosonde profile at 9:00 LT (Fig. 8 b). Above the temperature inversion, the large scale flow was from north-west with a wind velocity of about 11 m s$^{-1}$. The strong vertical wind shear between the easterly lake breeze and the strong westerly large scale flow caused mechanically induced turbulence (Ri<0.25), which led to a downward mixing of warmer air into the layer between 350 and 900 m amsl between 9:00 and 13:00 LT (Fig. 8 a). An inversion of 2 K formed at around 550 m amsl and a secondary weak inversion at 1200 m amsl represented the former ABL top at 13:00 LT. Between the two inversions a layer with westerly flow and reduced wind velocities compared to the large-scale flow established (Fig. 8).

To understand the entire development in the valley the upstream conditions are also relevant. The analysis of the numerical weather forecast model COSMO-EU (Steppeler et al., 2003) showed the development of a near neutral convective boundary layer (CBL) over land in the morning. Through the westerly large-scale flow, caused by the shallow Persian Through, the evolution of a well defined Mediterranean sea breeze front could not be observed (Fig. 11 a). Neither a temperature decrease nor a humidity increase, the typical characteristics associated with a sea breeze front propagating inland, were observable over the coastal plains. The advected air masses were mixed within the CBL and no frontal structure of the MSB could be observed. In the afternoon at 14:00 LT the large scale near surface flow over the Mediterranean Sea strengthened to approximately 8 m s$^{-1}$ and advected more stratified moist and cool air towards the coastal plains with a moisture maximum below the CBL top (Fig. 11 b). The inversion layer inhibited the CBL evolution. Potential temperature in the CBL over the plains stagnated at around 302 K and the thickness of the CBL decreased from around 960 m at 11:00 LT (Fig. 11 a) to 620 m at 14:00 LT (Fig. 11 b). At the same time at the mountain ridge the CBL became warmer by 3 K and CBL height increased from 735 m agl to 910 m agl. This indicates that the maritime air mass did not reach up the slopes at that time. (Fig. 11 b).



### 3.3.2 Stage II: Development of the foehn (15:00-18:30 LT)

In the afternoon the temperature gradient between the air at the mountain ridge and in the valley changed sign at 15:00 LT (Fig. 9). At the same time wind direction at the slope changed from east to west, horizontal wind velocity suddenly increased, vertical wind velocity became negative and specific humidity dropped (Fig. 9). This indicates that the air mass at ridge height

started to descend into the valley. In the valley itself, the foehn was observed at the KITcube location at 16:30 LT with a near surface wind velocity of $5.5\,\mathrm{m\,s^{-1}}$ and a strong increase of TKE to $3\,\mathrm{m^2\,s^{-2}}$ (Fig. 9). The vertical extent of the foehn, calculated from lidar data, was 1500 m agl, with a mean wind velocity of 6.0 to $7.0\,\mathrm{m\,s^{-1}}$ (Fig. 5 e). The temperature profile of the radiosonde at 17:00 LT showed a near neutral layer up to 1250 m agl with 308 K which was the temperature of the elevated layer between 550 and 1100 m amsl at 15:00 LT (Fig. 8 a). This is another indicator that the air from the elevated layer descended

into the valley and replaced the local air mass. From 17:00 until 18:30 LT the density current had a layer height of about 1400 to 1050 m agl and the horizontal radial mean wind velocity was 6.0 to 6.5 $\mathrm{m\,s^{-1}}$ (Fig. 5 e).

### 3.3.3 Stage III: Intensification of the foehn (18:30-21:00 LT)

The COSMO-EU analysis showed that upstream, with decreasing solar radiation in the evening, convection weakened and

potential temperature decreased. The mean potential temperature in the CBL over the coastal plains was 300.6 K, and at ridge height it was 304.3 K at 17:00 LT (Fig. 11 c). A comparison of the near surface model results with station observations from the IMS at Jerusalem and Bet Dagan showed comparatively good agreement, however, the temperature decrease in the afternoon occurred too early at both stations in the model. A time shift of 1.5 h was observed which means that the upstream afternoon conditions described from the model results have to be shifted from 17:00 LT to 18:30 LT. At that time the boundary layer

inversion over the coastal plains and at the mountain ridge strengthened to 4.1 K and 5.0 K, respectively, and the mean wind speed within the ABL increased to $5.1\,\mathrm{m\,s^{-1}}$ near the coast and $7.7\,\mathrm{m\,s^{-1}}$ at the ridge. The CBL height decreased considerably to 360 m agl over the plains and 330 m agl over the mountain ridge (Fig. 11 c).

In the valley and at the slope the mean wind velocity of the foehn increased to about $9\,\mathrm{m\,s^{-1}}$ at 18:30 LT, and the height of the foehn decreased to 350 m agl (Fig. 5 e). Also the near-surface measurements showed an increase of the wind velocity and

turbulent kinetic energy (Fig. 9). The radiosonde profiles at 19:00 LT showed a strong temperature inversion of 3.3 K at around 270 m agl (-80 m amsl) and the potential temperature in the layer was 305.6 K (Fig. 8 a). The profile of the horizontal wind had a clear jet like structure (Fig.8 b). Around 19:00 LT a sudden increase of the layer height and a reverse flow near the surface were detected east of the lidar (Fig. 10 a). There, the height of the foehn layer increased to approximately 1000 m amsl and the air below was quite turbulent (see also animation in the supplement). The backscatter RHI scan shows that the air mass in

the reverse flow has the same aerosol content as the air mass of the foehn layer, indicated by the same backscatter values of -7.7 dB. The air mass above the foehn layer has much lower backscatter values of -8.25 dB (Fig. 10 b). The high backscatter value of the foehn layer results most likely from local near surface dust emissions, caused by the high wind velocities and the increased turbulence.





The sudden increase of the layer height together with the near surface reverse flow and increased turbulence indicates a hydraulic jump with a rotor formation beneath. This phenomena can be explained by using the reduced-gravity shallow-water theory. It results when a subcritical flow upstream changes to a supercritical flow downstream. Then a sudden transition of the supercritical flow back to a subcritical state occurs by converting some of the flow's kinetic energy into an increase in potential

energy, i.e. a hydraulic jump. Calculating the Froude numbers for the coastal plain, mountain ridge, and valley conditions results in $Fr_{plains} = 0.73$, $Fr_{ridge} = 1.06$, and $Fr_{valley} = 1.7$, which confirms the assumption of a hydraulic jump in the valley. The reverse flow of the rotor causes the formation of a convergence line near the surface (Fig. 10 c). However this convergence line is not symmetrically aligned with the mountains. North of the lidar, the convergence line is located close to the shoreline of the DS, whereas south-east of the lidar location it is much closer to the measurement site. The non-uniform distance from

the mountains results most likely from the specific local orographic characteristics west of the lidar. North-west of the lidar a wadi opens into the Dead Sea valley. The westerly flow is most likely channelled in the wadi, resulting in accelerated wind speeds, and thus a stronger penetration into the valley, locating the convergence line further east. The convergence line was not stationary but retreated towards the western slope, whereby the rotor grew horizontally and vertically over the course of the event. On 21:00 LT the convergence line was observed at the KITcube location, with maximum TKE values of $4.6\,\mathrm{m^2\,s^{-2}}$. As

it further progressed towards the slope the TKE and wind velocity dropped at the KITcube location (Fig. 9). The foehn ceased around 21:30 LT, caused by further stratification of the upstream air masses, which led to a blocking of the flow at the Judean mountains, indicated by a non-dimensional mountain height of about 2.7 at 20:00 LT in the model, meaning about 21:30 LT in reality.

## 4   Summary and Conclusion

This study investigated frequently occurring foehn events in the DS valley using long-term near surface observations from 2014 until 2016, as well as high-resolution lidar measurements which were performed from these foehn events for the first time in 2014. With the automatic and probabilistic statistical mixture model from Plavcan et al. (2014) an objective identification and separation of foehn from the radiative-driven downslope flows was possible. The results confirmed the findings from earlier studies (e.g. Bitan, 1974; Alpert et al., 1982; Naor et al., 2017) that throughout the year, foehn evolves in the evening around

sunset. The new results reveal a higher occurrence frequency at the slopes than at the valley floor, indicating that the foehn penetrates downslope most of the days, but does not reach the valley floor in all occasions. We assume that the foehn detaches from the slope at the level of neutral buoyancy in the valley and continues flowing in an elevated layer above the valley floor. In the lidar data these elevated layers were observable and in one case also radiosonde ascents were available, which showed a low level inversion in the valley, strengthening our assumptions. These events are similar to cases in the Owens valley described by

Mayr and Armi (2010), where gap flows also detached from the slope at the level of neutral buoyancy.

Two main types of foehn in the DS valley could be identified with these new data. Weak and rather short events and strong, longer-lasting events which affect the whole valley, leading to a complete air mass exchange. In contrast to major downslope windstorms in other regions of the world (e.g Brinkmann, 1974; Mobbs et al., 2005; Grubišic et al., 2008; Gohm et al., 2008)





foehn at the DS occurs on a diurnal scale and only for a few hours per evening. This already shows that it is not synoptically driven but depends on diurnal local and mesoscale processes. In the presented case of a strong foehn event, the delayed cooling of the valley ABL in the afternoon led to the initiation of the foehn which was then intensified by supercritical flow conditions leading to a hydraulic jump and a rotor formation in the valley. A conceptual model was developed to summarise and illustrate the temporal and spatial interactions of the responsible mechanisms for such an event, based on the results of the 16 August 2014 (Fig. 12). The model was divided into 4 steps:

**Morning:** A strong temperature inversion decoupled the valley ABL from the large scale flow and thermally driven upslope winds and a lake breeze developed in the valley after sunrise (Fig. 12). A strong wind shear between the easterly lake breeze in the valley and the strong westerly large scale flow produced turbulence at the top of the valley ABL, indicated by a Richardson number below 0.25 (striped area in Fig. 12 a and b). Due to the turbulence, warmer air from above was mixed into the upper part of the valley ABL, and a secondary temperature inversion below mountain ridge height evolved. Even though the turbulent mixing decreased the inversion height, an undisturbed valley ABL existed below the inversion.

**Midday:** Between the two temperature inversions below and above ridge height a westerly flow with reduced wind speed formed a layer between the valley ABL with its easterly lake breeze and uplsope winds and the large-scale strong westerly flow. The valley itself was further heated by radiative forcing, which was concentrated within the reduced air volume in the valley below the lower inversion. The topographic amplification factor (Wagner, 1932; Steinacker, 1984) strengthened the valley heating additionally.

**Afternoon:** At the ridge cooling set in and eroded the inversion below ridge height. The cooler air from the flowing layer formed a deep foehn flowing down the slope. While descending into the valley the wind velocity accelerated. A change in the upstream conditions triggered the further development. The upstream subcritical flow reached a critical stage around ridge height, as indicated by the calculated Froude numbers of 0.73 over the coastal plains and 1.06 at the mountain ridge.

**Evening:** This transition upstream led to supercritical flow conditions downstream in the valley ($Fr = 1.7$). The layer height of the foehn in the valley decreased strongly and wind velocity increased respectively. In the valley the flow went from supercritical to a subcritical state forming a hydraulic jump with a rotor beneath. The layer height increased suddenly and mean wind velocities decreased generating severe turbulence. The rotor led to the formation of a convergence line near the ground. Finally, blocking of the upstream air masses by the mountains stopped the foehn as indicated by a non-dimensional mountain height of 2.7.

This study showed that foehn at the DS is initiated by the horizontal variation in boundary layer temperature across the mountain range. This is caused by an amplified heating of the valley ABL together with a delayed ABL development in the DS valley compared to the upstream boundary layer, as presented in this study. It can also be caused by the arrival of the cool MSB at the western side of the mountains as shown by various other studies (Ashbel and Brooks, 1939; Bitan, 1974; Segal et al., 1985; Lensky and Dayan, 2012; Naor et al., 2017).

The obtained results of this study are highly relevant for the DS region. They show that the foehn events are variable concerning their strength, layer height, and penetration into the valley. This is in particular important for the temperature, humidity, and aerosol distribution, in the valley, as well as for the evaporation from the DS water surface. Especially the coupling to evapo-





ration is of high importance as evaporation is the main loss of water from the DS (Metzger et al., 2018) determining the rate of the DS shrinking. We conclude that for correct climate, weather, and also evaporation forecasts in the region it is therefore important to correctly represent the boundary layer processes in the valley, in particular the inversion heights and the diurnal valley heating, as well as the occurring thermal wind systems, the ABL development upstream and the interaction with the

MSB. In the light of the ongoing climate change, and the shrinking of the DS, daily maximum temperatures in the valley could further increase making a penetration of foehn down into the valley more likely in the future. This of course would further increase evaporation, and the shrinking of the DS, resulting in a positive feedback cycle and change the diurnal temperature and humidity range.

These results also contribute to the wider understanding of the boundary layer processes in smaller valleys under weak syn-

optic forcing, where thermally driven local and mesoscale wind systems govern the wind field. Thermally driven foehn events similar to the ones in the DS valley were also observed in other areas of the world. The Washoe Zephyr occurs at the eastern slope of the Sierra Nevadas and is driven by the asymmetric heating between the lower western side and the elevated semiarid Great Basin (Zhong et al., 2008). At the lee side of the Cascade Mountains the winds are triggered by the temperature gradient between the semi-desert area on the lee side and the marine air masses on the upstream side close to the Pacific Ocean coast

(Doran and Zhong, 1994) and a similar phenomena was found by Bossert (1997) for the Mexico City basin. Even though these wind systems differ in scale, the trigger mechanism is similar in all cases and they can have a large impact on the local aerosol concentration and dispersion as well as on the local climatic conditions.

*Competing interests.* The authors declare that they have no conflict of interest.

*Acknowledgements.* The current study was carried out in the framework of the DEad SEa Research VEnue (DESERVE) (http://www.deserve-

vi.net), an international project funded by the Helmholtz Association of German Research Centres as a Virtual Institute (VH-VI-527), their partners and own contributions. We acknowledge our project partners, Professor Pinhas Alpert and his working group from Tel Aviv University for their support, the National Parks Authority Masada, in particular Eitan Campbell, for providing the measurement location Furthermore, we acknowledge support by Deutsche Forschungsgemeinschaft and Open Access Publishing Fund of Karlsruhe Institute of Technology.





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



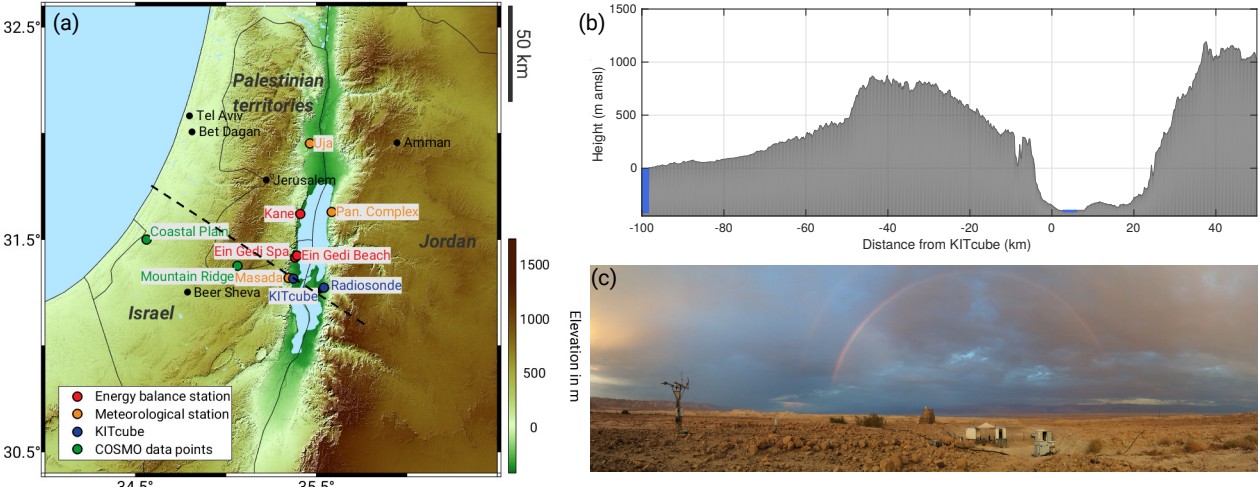

**Figure 1.** Topographic map of the research area with the measurement sites (a). Cross section along the dashed line in (a) is shown in (b), and a picture of the KITcube measurement site near Masada is shown in (c).

**Table 1.** Characteristics of the foehn events derived from RHI scans at 299° (dashed line in Fig. 1 a). Presented are mean wind speed, mean height, mean maximum wind speed, and mean height of the wind speed maximum of the foehn events. The mean wind speed was first calculated for each RHI scan and then averaged over the whole event. The height, maximum wind speed, and height of the wind speed maximum were determined for each RHI scan and then also averaged over the whole duration of the event. The events are sorted according to their duration.

| Date | mean wind speed (m s$^{-1}$) | mean layer height (m agl) | mean maximum wind speed (m s$^{-1}$) | mean height of maximum wind speed (m agl) | duration (hh:mm) | observed at KITcube location |
|---|---|---|---|---|---|---|
| 13.08.2014 | 3.4 | 420.0 | 5.1 | 199.0 | 1:51 | yes |
| 22.08.2014 | 4.1 | 529.1 | 5.6 | 293.6 | 2:09 | yes |
| 25.08.2014 | 6.2 | 501.8 | 8.3 | 312.7 | 2:39 | no |
| 14.08.2014 | 3.0 | 600.0 | 3.1 | 407.4 | 3:14 | no |
| 26.08.2014 | 7.2 | 961.1 | 10.8 | 406.8 | 4:13 | yes |
| 15.08.2014 | 7.5 | 1338.5 | 10.1 | 574.8 | 5:12 | yes |
| 16.08.2014 | 6.8 | 924.7 | 8.7 | 493.0 | 5:18 | yes |
| 18.08.2014 | 5.1 | 834.4 | 7.5 | 450.9 | 5:24 | yes |
| 28.08.2014 | 5.4 | 386.5 | 8.3 | 201.8 | 11:47 | yes |
| 31.08.2014 | 4.0 | 251.0 | 6.1 | 124.0 | 11:41 | yes |



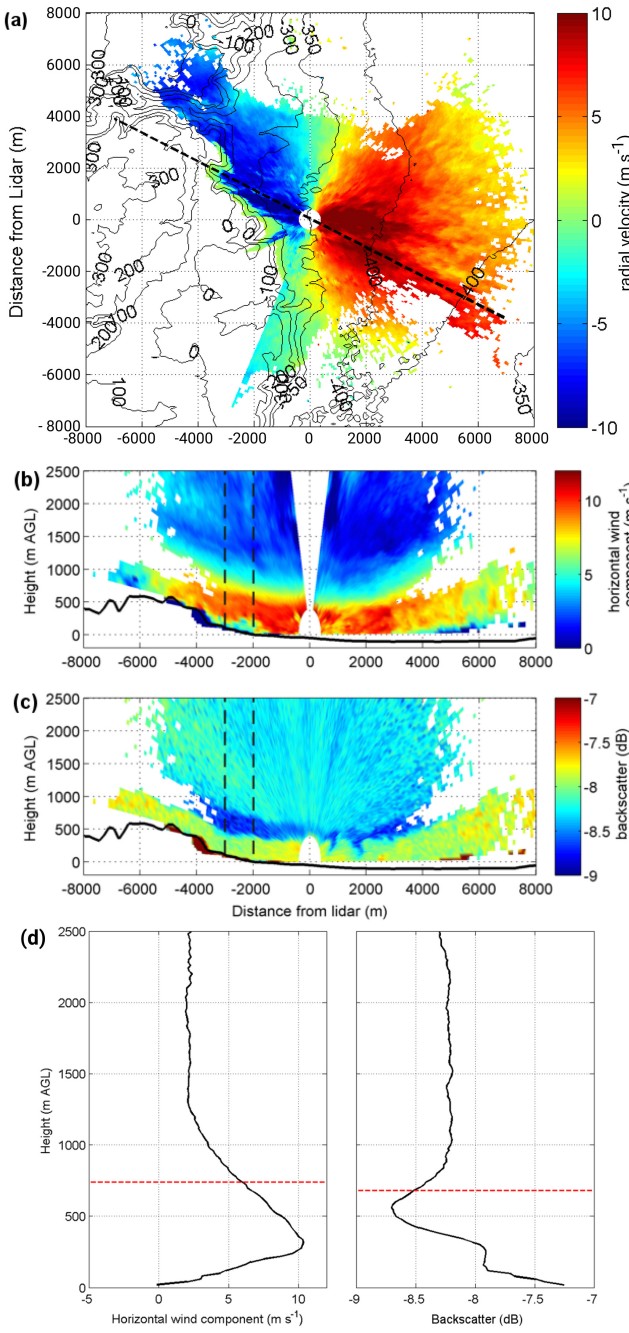

**Figure 2.** PPI scan of the radial velocity at 5° elevation on 16 August 2014, 19:01 LT (a). Blue colours indicate flow towards the lidar and red colours flow away from the lidar. The dashed line indicates the direction of the RHI scans of the horizontal component of the radial velocity shown in (b) and of the backscatter signal shown in (c) (note the different colour scales). RHI scans were performed on 16 August 2014, 19:06 LT. Vertical profiles of the horizontal radial wind velocity component and of the backscatter signal (d) averaged between the black dashed lines in (c). The red dashed lines indicate the foehn layer heights, automatically detected by the strongest vertical gradient of the respective profile above the maximum.



**Figure 3.** Left: Probability of foehn (colors) for wind velocity and temperature difference ($\Delta T = T_{crest} - T_{valley}$). Right: Relative frequency of foehn occurrence with a detection probability of more than 75 %. Sunrise and Sunset are marked as red lines.





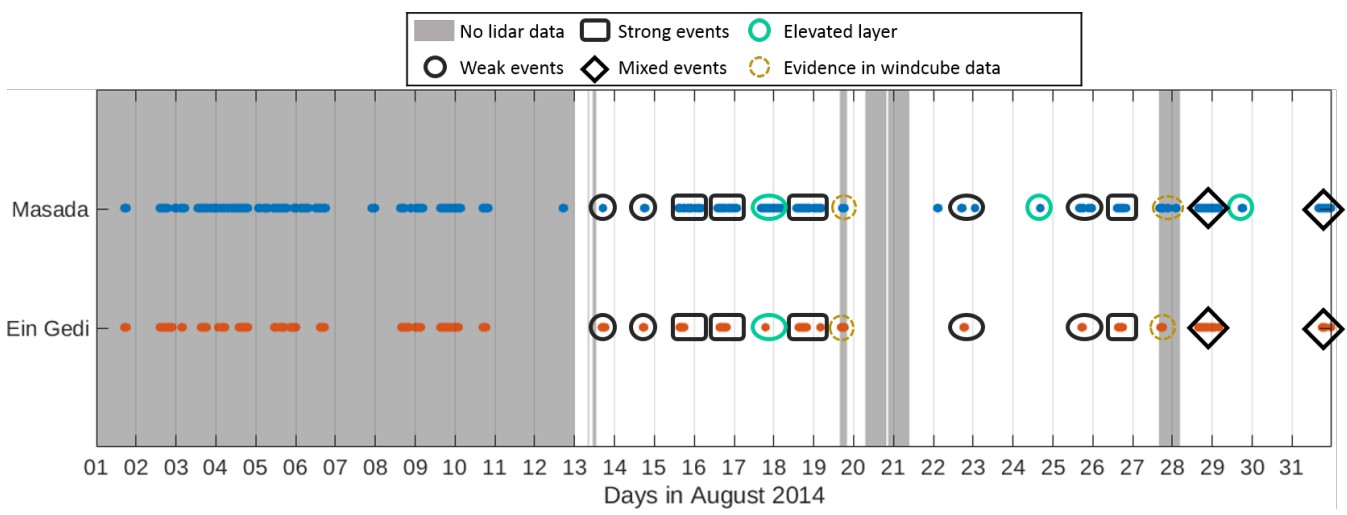

**Figure 4.** Occurrence of foehn in August 2014 at the slope (Masada, blue dots) and in the valley (Ein Gedi, red dots). Shown is the foehn occurrence with a detection probability of more than 75 %. Grey shaded area shows the dates were lidar data were not available.





**Figure 5.** Example for a short and weak foehn event (a,b,c) and for a strong foehn event (d,e,f). Shown are 10 min mean vertical profiles of horizontal wind at the KITcube location (a,d). Length and colours of the arrows represent wind velocity and the direction of the arrows indicate the horizontal wind direction. Time-height cross sections of the averaged vertical profiles of the horizontal radial wind component derived from lidar RHI scans at 299°(b,e). Negative values (blue colours) indicate a wind component from north-west and positive values (red colours) indicate a wind component from south-east. Black stars indicate the detected height of the foehn. Below the cross sections the variance of the vertical wind ($\sigma_w$) at 2 m agl and wind direction ($wd$) measurements in 40 m height are shown. Hovmoeller diagrams of the near-surface radial velocity from lidar RHI scans at an azimuth angle of 299° and 7° elevation west of the lidar and 1.6° east of the lidar (c,f). The black dashed line indicates the end of the western slope and the beginning of the valley floor. Wind direction ($wd$) was measured in 40 m agl. Orange points highlight wind directions between 270 and 330°, the main wind direction of the foehn.





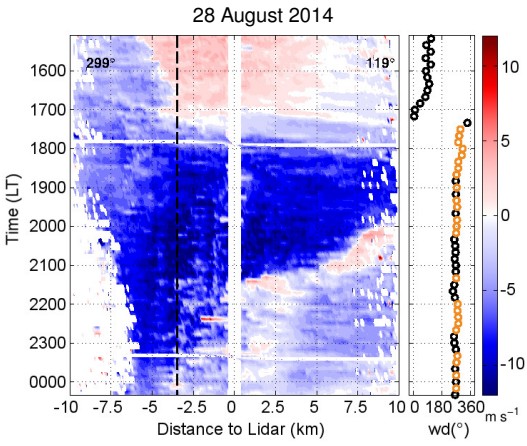

**Figure 6.** Example for a mixed event on 28 August 2014. Hovmoeller diagram of the near-surface radial velocity from lidar RHI scans at an azimuth angle of 299° and 7° elevation west of the lidar and 1.6° east of the lidar. The black dashed line indicates the end of the western slope and the beginning of the valley floor. Wind direction ($wd$) was measured in 40 m agl. Orange points highlight wind directions between 270 and 330°, the main wind direction of the foehn.

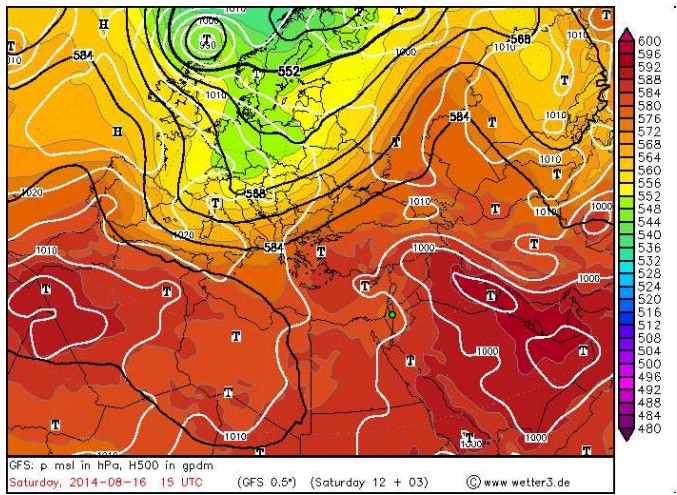

**Figure 7.** Large scale synoptic conditions on 16 August 2014, 15 UTC. Geopotential in gpdm at 500 hPa (black contours), surface pressure in hPa (white contours), relative topography 500/1000 hPa (coloured). Green dot indicates investigation area.



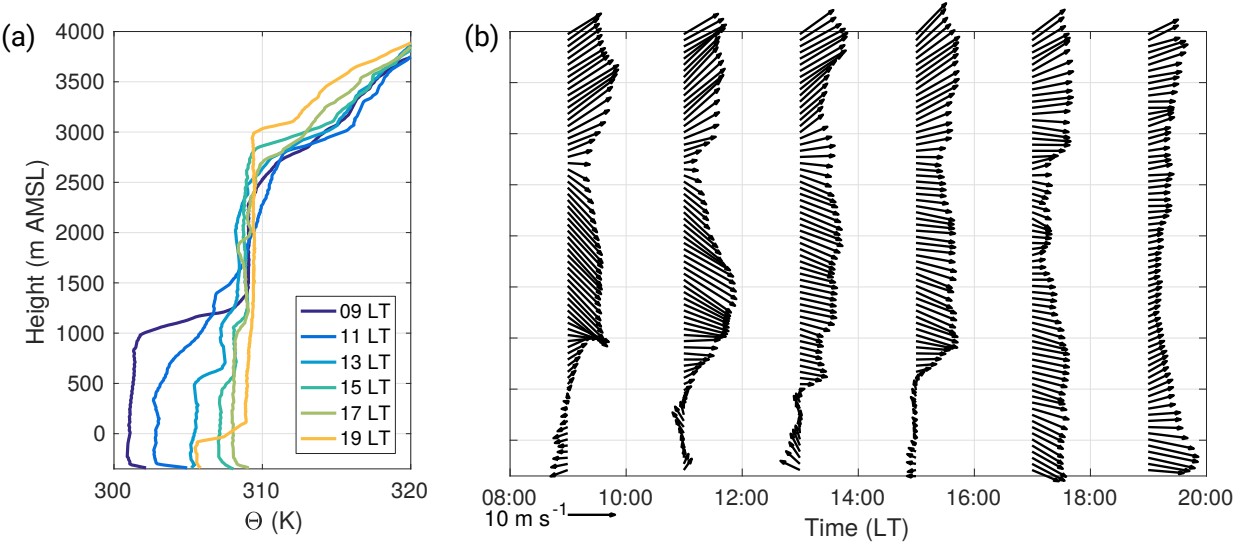

**Figure 8.** Potential temperature (a) and horizontal wind profiles (b), measured with radiosondes in the valley at the KITcube location. Data are shown for 16 August 2014.





**Figure 9.** Time series of potential temperature ($\Theta$), potential temperature difference between crest and slope/valley ($\Delta\Theta$), wind direction ($wd$), wind speed ($ws$), vertical wind speed ($w$), specific humidity ($q$), and turbulent kinetic energy (TKE) for Jerusalem (crest), Masada (slope), and KITcube location (valley) on 16 August 2014.





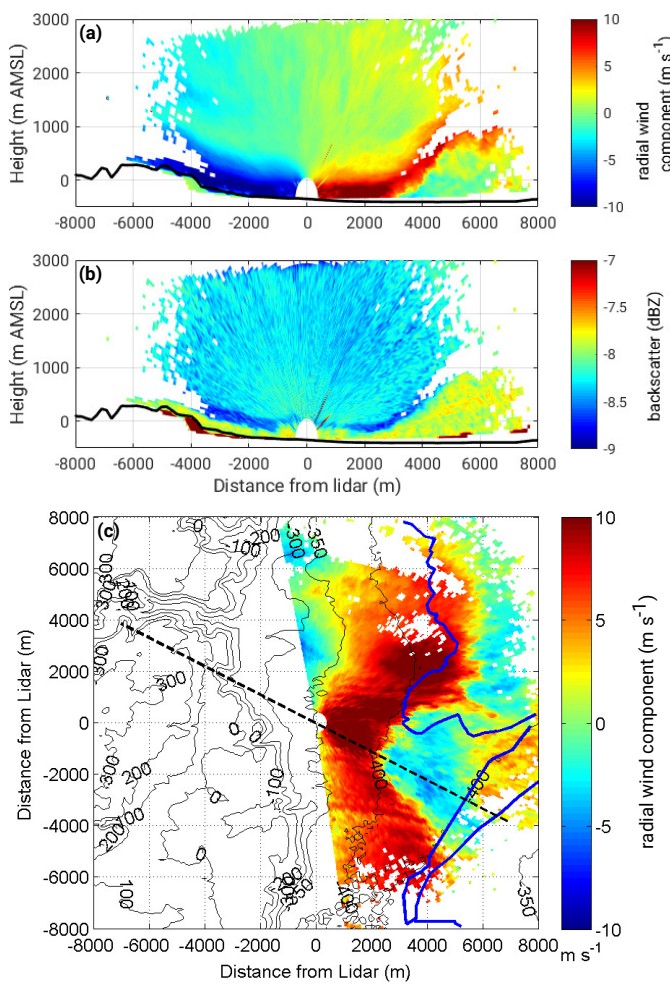

**Figure 10.** RHI scan at an azimuth angle of 299° (dashed line in (c)) showing radial velocity (a) and backscatter (b) on 16 August 2014, 19:36 LT. PPI scan at an elevation angle of 0.2°, shows radial velocity (c) at 19:51 LT. In (a) and (c) blue colours indicate a flow towards the lidar and red colours indicate a flow away from the lidar. The shoreline of the Dead Sea is indicated by a dark blue line in (c).



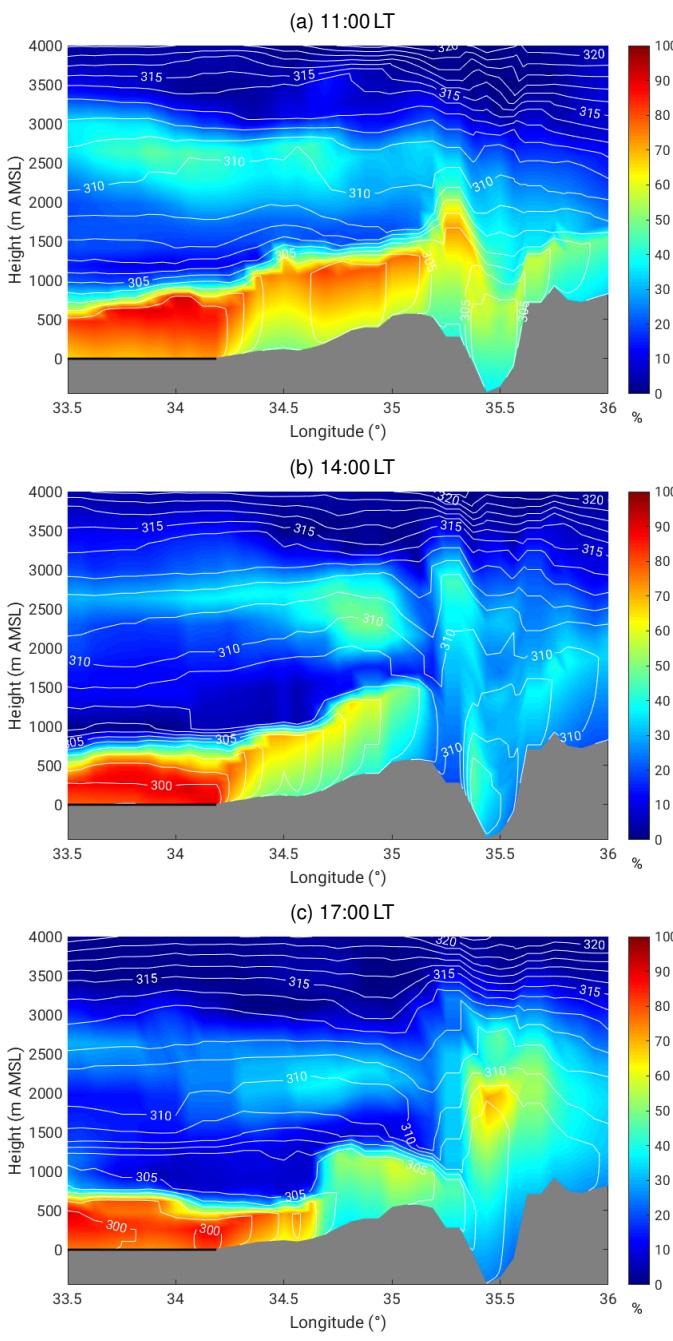

**Figure 11.** West-east cross section of relative humidity (coloured) and potential temperature (isolines) at the latitude of Masada (31.3125°N) from COSMO-EU analysis data for 16 August 2014.





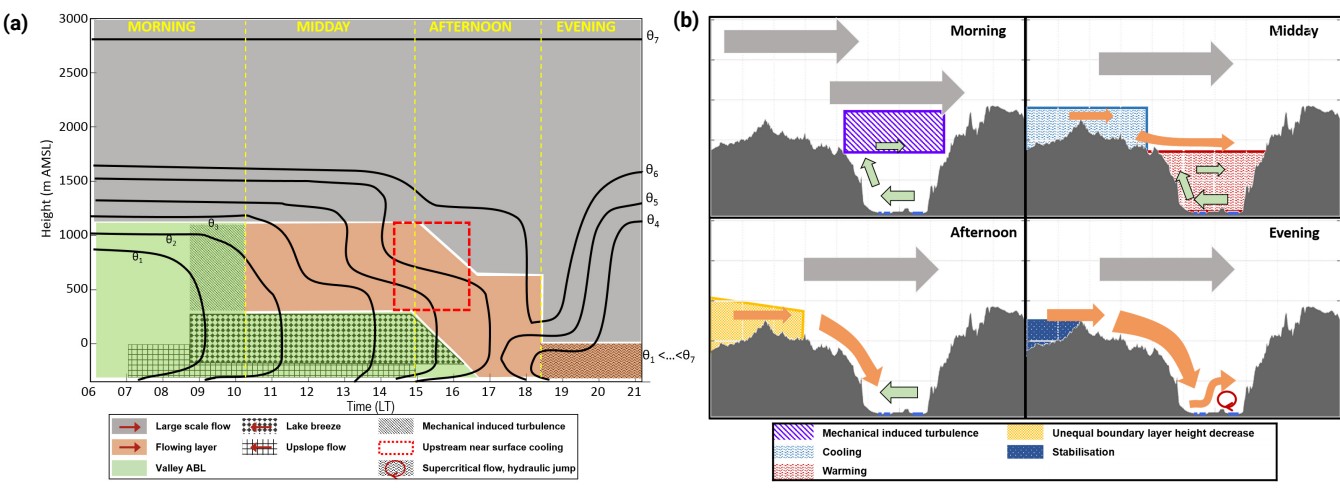

**Figure 12.** Conceptual model describing the different boundary layer processes in the valley on 16 August 2014. The temporal evolution of the different layers and wind systems is given in (a). The wind systems (arrows) and the ongoing processes (shaded areas) leading to the next stage for morning, noon, afternoon, and evening, are shown in (b).