# Peer review of "Characteristics and evolution of diurnal foehn events in the Dead Sea valley"

_Atmospheric Chemistry and Physics, 2018_

## Referee Comment (RC1) · Anonymous Referee #1 · 9 Aug 2018

Review of: Characteristics and evolution of diurnal foehn events in the Dead Sea valley I found this paper very important and interesting particularly the unique observations done in the DS area and described hear. I recommend to accept the paper after making a revision in light of the comments below. P1l6 (page 1 line 6) "the mean maximum velocities of around 5m/s". Mean of the maximum is an exact number please give the exact value. Section 3.1 It will be helpful if you will add some sentences describing the main differences between density and radiative driven flow, and their relation to potential temperature difference between the crest and bottom. What are the problems that the automatic mixture model deal with. p.6 line 5. The distinction between foehn and radiative flow are determined mainly by the potential temperature difference between the crest, Jerusalem 810 m and Masada, -7m p6l9 or Ein Gedi -427m p6l10. In Fig. 3

the temperature differences are shown. Please show the potential temperature differences instead. Also it looks like a mistake the positive temperature difference between Jerusalem and Ein Gedi, p6l13, Jerusalem is always cooler than Ein Gedi.

P7l16-l19 The west wind observed at least 2 hours earlier at 14:00. The height of the maximum west wind at 14:00 is at 1750m, and so at 16:00. p8l19-26 The mixed events. At least in August 28 it is not clear to me why you call it mixed event and August 16 strong event. Comparing Fig. 5f after 20:00 and Fig. 6 after 21:00 the wind behavior is similar and so both cases can considered as mixed events. P9l10 Please note that from the ground up to 900m the stratification is unstable or neutral i.e. The strong vertical mixing from the ground to 900m discussed in, P9l18-20, is mainly due to the unstable layer close to ground, and not due to mechanical mixing. P9l16 It is not clear that the western wind is the return flow. I think it is the residual wind from the night. P9l23-34 What was the resolution in COSMO-EU model, may be the not observed front is a consequence of the resolution. The flow field is not shown and could be very helpful in interpret the model results since the wind field play a major role in understanding the foehn. I recommend to add a figure o the flow field. Please also show the mixing ratio (q) instead RH fields in Fig. 11, since q is more relevant. p10-11 Stage III. You combine observations, model results and interpretation, I suggest to note what based on observations what based on model and what is interpretation. I also suggest to add a model wind cross section at 21:00. The description of the hydraulic jump you refer looks like a sea breeze front/ gravity current head. You did not address one of the main questions: is this sea breeze front developed over the plains much earlier and propagates into the DS, or it developed in the valley. The formation of the foehn is not clear, since the authors claim that the source of the air is the Mediterranean and we would expect high q of this sea air relative to the inland air. Please explain the formation of the dry air. It is not clear if the hydraulic jump observed in the DS is formed there or maybe it is the sea breeze front formed over the plains and mountains and propagates downslope into DS. If the hydraulic jump is the sea breeze front, this might explains the elevated foehn found in some cases since under certain conditions the front leave

the mountain and propagates on isopycnicals. P12l7-12 The mixing is mainly due to convection from the unstable layer near the ground see my comment P9l10 above.
* * *

---

## Referee Comment (RC2) · Anonymous Referee #2 · 16 Aug 2018

Review of the paper "Characteristics and evolution of diurnal foehn events in the Dead Sea valley", by J. Vüllers, G.J. Mayr, U. Corsmeier and C. Kottmeier.

This paper deals with the statistical and dynamical characteristics of an interesting diurnal mesoscale phenomenon (namely the foehn that sometimes is mentioned even in the non-specialist literature of the area). This paper is well written and the mesoscale analysis, including a detailed description of the different stages of the phenomenon, is rather convincing. Therefore, I consider this paper as worth of publication in Atmospheric Chemistry and Physics, but with a minor revision, taking into account my comments below.

Page 5, section 2.5: this section should be converted into an appendix (of course keeping here only the definition of symbols used below), because it contains just a summary of the reduced-gravity theory of shallow flow over obstacles (to be referenced below in paper), with no original aspects.

Page 6, from line 6 to line 10: the Jerusalem temperature is used as representative of T at the "crest". However, Jerusalem is located at about 50 km north of the cross-section of Fig. 1. Moreover, in the same sentence a "downstream station" is mentioned with no additional specification. Below, the Masada station is probably identified as such downstream station. The entire paragraph is rather involved and needs better explanation/phrasing.

Page 6, line 13: please specify the temperature differences (T crest minus T valley?).

Page 6, lines 13-15: this sentence is unclear. Most probably, "were" should be "where", but even with this correction, still the sentence needs to be improved a little.

Page 8, line 11: please refer to Fig. 1 for the radiosonde location. Moreover, "the other side" is ambiguous – it is probably the eastern side of the DS: please clarify.

Page 9, lines 20-21 (and somewhere else): here the word "inversion" refers to the profile of potential temperature $\Theta$ (fig. 8). However, normally the word inversion is used to denote temperature T increasing with height. It is not obvious if the stable layer of fig. 8 implies an increase of T with height. There is an ambiguity across the paper in the use of the word "inversion" that should be avoided unless a real "T inversion" is implied. A similar ambiguity is also in the use of "warmer" or "cooler": such words should refer only to T and not to $\Theta$.

Page 9, line 26: Fig. 11 is introduced here, while Fig. 10 is referenced only below in sect. 3.3.3 for the first time. This should be avoided: I think that figures should be numbered in the order of citation.

Page 10, line 18: any hint for the cause of the earlier cooling in the COSMO model?

Page 12, lines 1-3: "depends on diurnal local and mesoscale processes": please try to be more specific - for instance the MSB is mentioned below (line 30) as the main cause of the westerly flow from which the foehn takes its energy. However, in sect. 3.3 a synoptic-scale pressure gradient is invoked as being important, at least for the strongest cases. I think it is not made clear enough to what extent the MSB alone is sufficient to initiate the DS foehn.

Page 12, line 18: is "the ridge cooling" due to radiation or also to cold air advection from the Mediterranean (arrival of MSB)?

Page 13, lines 5-7: however, in a warming scenario, temperature may increase also upstream and not only within the valley, so the impact on T profile is not obvious (or perhaps it is implicitly assumed that, the MSB being important, the Mediterranean sea temperature will increases more slowly that the continental temperature?).

Figures:

Fig. 1: perhaps the left panel (the map) should be enlarged.

Fig. 3: please specify in the captions where crest and valley temperatures are measured, respectively (or refer precisely to the text where this is explained).

Fig. 12 a: this is a laudable attempt to synthesize a conceptual model in a picture. However, it is difficult to appreciate the different hatchings in the green area, unless one enlarges the page on a (large) screen. Moreover, the small rectangles in the inset below (the legend) are not sufficiently clear.

Typos in the text:

p. 2, line 4: depend.

p.2, line 9: "and MAP" in place of "or MAP".

p. 5, line 15: drop comma after "hereby".

---

## Author Comment (AC1) · 7 Nov 2018

**Reply to Reviewer 1**

Jutta Vüllers et al.
(jutta.vuellers@kit.edu)

**1 General Comments**

*I found this paper very important and interesting particularly the unique observations done in the DS area and described hear. I recommend to accept the paper after making a revision in light of the comments below.*

Thank you for the very insightful review. Your comments helped to improve the paper. Responses to individual comments are provided below. Reviewer's comments are in italic.

**2 Specific Comments**

*C1 - P1 l6 (page 1 line 6) "the mean maximum velocities of around 5m/s". Mean of the maximum is an exact number please give the exact value.*
A) The sentence was changed to:
T) Type I has a duration of approximately 2-3 h and a mean maximum velocity of 5.5 m s$^{-1}$ and does not propagate far into the valley, whereas type II affects the whole valley, as it propagates across the valley to the eastern side. Type II reaches mean maximum wind velocities of 11 m s$^{-1}$ and has a duration of about 4-5 h.

*C2 - Section 3.1 It will be helpful if you will add some sentences describing the main differences between density and radiative driven flow, and their relation to potential temperature difference between the crest and bottom. What are the problems that the automatic mixture model deal with.*
A) A short description of the differences and the relation to the potential temperature was added.
T) Density-driven flows are possible when the potential temperature of the upstream air mass crossing at crest height is equal or lower than the temperature in the valley. The air mass descends resulting in similar potential temperatures at the crest and in the valley. In contrast, radiative-driven downslope flows are triggered by radiative cooling of the air layer near the slope and the resulting temperature gradient between the air at the slope and the air at the same height in the valley centre (Whiteman, 2000). This leads to a stable stratification and therefore to a positive potential temperature difference between the crest and the valley ($\Delta\Theta = \Theta_{crest} - \Theta_{valley} > 0$).

*C3 - p.6 line 5. The distinction between foehn and radiative flow are determined mainly by the potential temperature difference between the crest, Jerusalem 810 m and Masada, -7m p6l9 or Ein Gedi -427m p6l10. In Fig. 3 the temperature differences are shown. Please show the potential temperature differences instead. Also it looks like a mistake the positive temperature difference between Jerusalem and Ein Gedi, p6l13, Jerusalem is always cooler than Ein Gedi.*
A) Thank you for this comment. The figure already shows the potential temperature difference, only the axes and the description was wrong. The potential temperature difference shown in the figure is $\Delta\Theta = \Theta_{crest} - \Theta_{valley}$. Thus, positive values mean potential temperature in Jerusalem is higher than Ein Gedi, in particular during radiatively-driven downslope flows, where a stable stratification occurs.

*C4 - P7l16-l19 The west wind observed at least 2 hours earlier at 14:00. The height of the maximum west wind at 14:00 is at 1750m, and so at 16:00.*
A) The description was not precise enough. What we meant is, that it started to penetrate into the valley at

16:00 LT. The text was changed.

T) Before the west wind reached the valley, an elevated west wind maximum just above crest height (1750 m agl) was observed  in the afternoon. At 16:00 LT,  it started to penetrate down into the valley  and reached the valley floor at around 18:30 LT.

*C5 - p8l19-26 The mixed events. At least in August 28 it is not clear to me why you call it mixed event and August 16 strong event. Comparing Fig. 5f after 20:00 and Fig. 6 after 21:00 the wind behaviour is similar and so both cases can considered as mixed events.*

A) The difference is that in the case of the 16 August the foehn ceases at 21:00 completely. There is no foehn layer with the typical characteristics, such as the jet like structure observed anymore. Not in the valley, and also not at the slopes. In contrast, on August 28 the foehn first reaches far into the valley and after 21:00 it continues to be present with a well defined jet like structure, but only at the slope and at the foot of the slopes for nearly another 6 h (compare also Table 1 with the durations of the events). That's why we call August 28 a 'mixed event'. There are changes of the penetration distance into the valley and a change in intensity, but it keeps the typical characteristics of a well defined layer with a jet like structure for a total of 11:47h.

*C6 - P9l10 Please note that from the ground up to 900m the stratification is unstable or neutral i.e. The strong vertical mixing from the ground to 900m discussed in, P9l18-20, is mainly due to the unstable layer close to ground, and not due to mechanical mixing.*

A) We agree that the atmosphere is close to neutral at 9 LT and that there is of course convection and strong vertical mixing from the ground, but in our opinion strong vertical mixing from the ground would not lead to a secondary inversion and a temperature increase only in the upper part, between 500 and 900 m AMSL, which was observed in this case. The formation of this layered structure with two inversions is in our opinion due to mechanically induced downward mixing (Ri<0.25) of warmer air into the upper part of the boundary layer.

*C7 - P9l16 It is not clear that the western wind is the return flow. I think it is the residual wind from the night.*

A) During the night (00:00 - 07:30 LT) no west wind is observed in the valley. The wind profiles derived from the lidar show only north-easterly to easterly winds up to a height of 1100 m AMSL. Therefore, we assume that the observed westerly winds between 500 and 900 m amsl are the return flow of the lake breeze and not a residual flow.

*C8 - P9l23-34 What was the resolution in COSMO-EU model, may be the not observed front is a consequence of the resolution. The flow field is not shown and could be very helpful in interpret the model results since the wind field play a major role in understanding the foehn. I recommend to add a figure o the flow field. Please also show the mixing ratio (q) instead RH fields in Fig. 11, since q is more relevant.*

A) The resolution of the model is 7 km. As you suggested we now show specific humidity instead of relative humidity. Furthermore, we decided against showing the 2D flow field as the large scale near surface wind field can also be seen in Fig. 7, from the position of the isobars. However, we show the u-component of the wind along the cross section as we hereby also have the vertical information of the wind. Here you can see that a separation of a possible sea breeze from the large scale flow is not possible.

*C9 - p10-11 Stage III. You combine observations, model results and interpretation, I suggest to note what based on observations what based on model and what is interpretation. I also suggest to add a model wind cross section at 21:00. The description of the hydraulic jump you refer looks like a sea breeze front/ gravity current head. You did not address one of the main questions: is this sea breeze front developed over the plains much earlier and propagates into the DS, or it developed in the valley. The formation of the foehn is not clear, since the authors claim that the source of the air is the Mediterranean and we would expect high q of this sea air relative to the inland air. Please explain the formation of the dry air. It is not clear if the hydraulic jump observed in the DS is formed there or maybe it is the sea breeze front formed over the plains*

*and mountains and propagates downslope into DS. If the hydraulic jump is the sea breeze front, this might explains the elevated foehn found in some cases since under certain conditions the front leave the mountain and propagates on isopycnicals.*

A) Concerning your first point of clarification what results come from the model and what from the measurements: We rephrased some parts to hopefully make it clearer.

Your second question was, whether a sea breeze front propagated into the DS or not. In the discussed case of 16 August a propagation of a clear sea breeze front into the DS can not be observed in the COSMO-EU model data. Neither in the wind field (see also new cross sections which are added to the paper), nor in the temperature or humidity data. With a sea breeze front development, I would expect to see a strong temperature and humidity gradient at the sea breeze front head, propagating over the coastal plains and transporting moist air towards the mountains and in the valley. However, this was not seen in the data (Fig. 10). Of course synoptic large scale flow from west transports some maritime air towards the land, but no well defined sea breeze front with clear frontal structure or head is observed.

The foehn is triggered in the evening by the radiative cooling at the crest, as this starts earlier than in the valley. The intensification of the foehn, which results in the hydraulic jump in the valley, comes from the different development of the boundary layer over the plains, but not from the entrance of the sea breeze front into the valley.

In other cases the foehn is certainly connected to the sea breeze front entering the valley, which was already shown in several other papers, but not on 16 August.

We added an additional figure with cross sections of the wind. However, for the time you suggested (21 LT) we do not have model data. We do have model results for model time 20 LT this would correspond to 21:30 LT in reality as the events were predicted in the model too early. At this time, the event already ended. Therefore, and most likely also due to limited model resolution of 7 km, the hydraulic jump can not be seen in the model data. T) At that time  model results show that the potential temperature at the CBL top increased by 4.1 K over the coastal plains and at the mountain ridge  by 5.0 K,  [...]  Measurements in the valley and at the slope show that the mean wind velocity of the foehn increased to about $9\,\mathrm{m\,s^{-1}}$ at 18:30 LT, and the height of the foehn decreased to 350 m agl (Fig.5 e).

There, the height of the foehn layer increased to approximately 1000 m amsl and the air below was quite turbulent (see also animation of lidar measurements in the supplement).

Calculating the Froude numbers for the coastal plain (from model), mountain ridge (from model), and valley  (from measurements) results in $Fr_{plains} = 0.73$, $Fr_{ridge} = 1.06$, and $Fr_{valley} = 1.7$, which confirms the assumption of a hydraulic jump in the valley.

*C10 - P12l7-12 The mixing is mainly due to convection from the unstable layer near the ground see my comment P9l10 above.*
A) please see answer to comment C6.

**References**

Whiteman, C. (2000). *Mountain Meteorology: fundamentals and applications.* Oxford Univ. Press, New York.

---

## Author Comment (AC2) · 7 Nov 2018

**Reply to Reviewer 2**

Jutta Vüllers et al.
(jutta.vuellers@kit.edu)

**1 General Comments**

*This paper deals with the statistical and dynamical characteristics of an interesting diurnal mesoscale phenomenon (namely the foehn that sometimes is mentioned even in the non-specialist literature of the area). This paper is well written and the mesoscale analysis, including a detailed description of the different stages of the phenomenon, is rather convincing. Therefore, I consider this paper as worth of publication in Atmospheric Chemistry and Physics, but with a minor revision, taking into account my comments below*

Thank you for the very insightful review. Your comments helped to improve the paper. Responses to individual comments are provided below. Reviewer's comments are in italic.

**2 Specific Comments**

*C1 - Page 5, section 2.5: this section should be converted into an appendix (of course keeping here only the definition of symbols used below), because it contains just a summary of the reduced-gravity theory of shallow flow over obstacles (to be referenced below in paper), with no original aspects.*
A) Thank you very much for your comment. You are right, this is just a summary of the reduced-gravity theory of shallow flow. However, we think that this explanation is helpful for the reader to fully understand the paper, as this theory is applied to explain the observed phenomena. We think that moving part of the section to the appendix, but keeping the symbols and definitions here, does not increase the readability of the paper.

*C2 - Page 6, from line 6 to line 10: the Jerusalem temperature is used as representative of T at the "crest". However, Jerusalem is located at about 50 km north of the cross-section of Fig. 1. Moreover, in the same sentence a "downstream station" is mentioned with no additional specification. Below, the Masada station is probably identified as such downstream station. The entire paragraph is rather involved and needs better explanation/phrasing.*
A) Thank you for this comment. We rephrased the paragraph to make clear which downstream stations are used. Regarding your concern about the distance of Jerusalem to the stations, we agree that we make the assumption that the conditions at the crest just above Ein Gedi are the sames as in Jerusalem. However, we think that the temperatures in Jerusalem are also representative for the crest 50km south, as the larger scale conditions are the same and landscape/vegetation are similar.
T) The wind regimes are identified using the  potential temperature difference ($\Delta\Theta$) between the crest (Jerusalem, 810 m amsl, Fig. 1 a) and two downstream stations, one at the slope (Masada, -7 m amsl, Fig. 1 a) and one in the valley (Ein Gedi Beach, -427 m amsl, Fig. 1 a), as well as the wind speed at the respective downstream station. The only parameter which has to be set prior to the fully automatic classification is the wind direction sector indicating "downslope".  For Masada, this is 200-315° and for Ein Gedi Beach  220-320°.

*C3 - Page 6, line 13: please specify the temperature differences (T crest minus T valley?).*
A) The potential temperature difference was specified in the text. It's $\Delta\Theta = \Theta_{crest} - \Theta_{valley}$
T) . This leads to a stable stratification and therefore to a positive potential temperature difference between the

crest and the valley $(\Delta\Theta = \Theta_{crest} - \Theta_{valley} > 0)$.

*C4 - Page 6, lines 13-15: this sentence is unclear. Most probably, "were" should be "where", but even with this correction, still the sentence needs to be improved a little.*
A) The sentence was rephrased
T) This coincides with literature values  of 1-5 m s$^{-1}$ for the maximum wind velocities of radiative driven downslope flows (Whiteman, 2000; Zardi and Whiteman, 2013).

*C5 - Page 8, line 11: please refer to Fig. 1 for the radiosonde location. Moreover, "the other side" is ambiguous – it is probably the eastern side of the DS: please clarify.*
A) The reference to Fig.1 was added and the sentence changed to clarify that the eastern side is meant with "other side".
T) However, radiosondes launched at the eastern shore of the DS (Fig. 1), indicate that the foehn reached over the DS towards the eastern shore.

*C6 - Page 9, lines 20-21 (and somewhere else): here the word "inversion" refers to the profile of potential temperature $\Theta$ (fig. 8). However, normally the word inversion is used to denote temperature $T$ increasing with height. It is not obvious if the stable layer of fig. 8 implies an increase of $T$ with height. There is an ambiguity across the paper in the use of the word "inversion" that should be avoided unless a real "T inversion" is implied. A similar ambiguity is also in the use of "warmer" or "cooler": such words should refer only to $T$ and not to $\Theta$.*
A)Thank you for this comment. In fact the stable layers discussed in this paper are all connected to temperature inversions, meaning an increase of temperature with height. However, the temperature profiles are not shown additionally to the potential temperature profiles. As we only show potential temperature in the paper we revised the paper and rephrased the critical sentences.

(i)In the morning a strong temperature inversion, marked the height of the  valley ABL at 900 m amsl, resulting in a well defined capping stable layer (Fig. 8 a).

(ii)The strong vertical wind shear between the easterly lake breeze and the strong westerly large scale flow caused mechanically induced turbulence (Ri¡0.25), which led to a downward mixing of warmer air into the layer between 350 and 900 m amsl between 9:00 and 13:00 LT, which also increased potential temperature (Fig. 8 a).  A temperature inversion formed at around 550 m amsl, resulting in a stable layer with a potential temperature increase of 2 K. A secondary weaker inversion at 1200 m amsl represented the former ABL top at 13:00 LT.

(iii)At the same time at the mountain ridge the potential temperature of the CBL increased by 3 K and CBL height increased from 735 m agl to 910 m agl.

(iv)At that time  model results show that the potential temperature at the CBL top increased by 4.1 K over the coastal plains and at the mountain ridge  by 5.0 K. The mean wind speed within the ABL increased to 5.1 m s$^{-1}$ near the coast and 7.7 m s$^{-1}$ at the ridge.

*C7 - Page 9, line 26: Fig. 11 is introduced here, while Fig. 10 is referenced only below in sect. 3.3.3 for the first time. This should be avoided: I think that figures should be numbered in the order of citation.*
A) Thank you for pointing that out. Indeed we agree that figures should be numbered in order of citation. We changed the figure order accordingly.

*C8 - Page 10, line 18: any hint for the cause of the earlier cooling in the COSMO model?*
A) We assume that this earlier cooling is just introduced through the coarse time resolution of the output data, which forced us to interpolate between the output time steps. Output was only available every 3 h.

*C9 - Page 12, lines 1-3: "depends on diurnal local and mesoscale processes": please try to be more specific - for instance the MSB is mentioned below (line 30) as the main cause of the westerly flow from which the foehn takes its energy. However, in sect. 3.3 a synoptic-scale pressure gradient is invoked as being important, at least for the strongest cases. I think it is not made clear enough to what extent the MSB alone is sufficient to initiate the DS foehn.*

A) We rephrased the sentence being more specific.

T) This already shows that it is not synoptically driven but depends on diurnal local and mesoscale processes , such as the delayed development and cooling of the valley ABL or the MSB reaching the Judean mountains in the afternoon, both leading to a horizontal temperature gradient across the mountain range.

*C10 - Page 12, line 18: is "the ridge cooling" due to radiation or also to cold air advection from the Mediterranean (arrival of MSB)?*

A) this cooling was caused by radiative cooling. In this case the MSB did not reach the ridge (see also Fig 10c).

T) At the ridge radiative cooling set in and eroded the inversion below ridge height.

*C11 -Page 13, lines 5-7: however, in a warming scenario, temperature may increase also upstream and not only within the valley, so the impact on T profile is not obvious (or perhaps it is implicitly assumed that, the MSB being important, the Mediterranean sea temperature will increases more slowly that the continental temperature?).*

A) Concerning the warming scenario it is indeed assumed that the Mediterranean Sea temperatures will increase more slowly than continental temperatures, and the second point refers to the shrinking of the DS a shrinking water surface will result in increased sensible heat flux and thus increased temperatures in the valley.

*C12 -Fig. 1: perhaps the left panel (the map) should be enlarged.*

A) We have enlarged the map.

*C13 -PFig. 3: please specify in the captions where crest and valley temperatures are measured, respectively (or refer precisely to the text where this is explained).*

A)The caption was changed.

T) Left: Probability of foehn (colors) for wind velocity and potential temperature difference ($\Delta T = T_{crest} - T_{valley}$ $\Delta \Theta = \Theta_{crest} - \Theta_{valley}$). The crest station is Jerusalem and the valley stations are Masada and Ein Gedi (Fig. 1). Right: Relative frequency of foehn occurrence with a detection probability of more than 75 %. Sunrise and Sunset are marked as red lines.

*C14 -Fig. 12 a: this is a laudable attempt to synthesize a conceptual model in a picture. However, it is difficult to appreciate the different hatchings in the green area, unless one enlarges the page on a (large) screen. Moreover, the small rectangles in the inset below (the legend) are not sufficiently clear.*

A) Thank you for this remark. We enlarged the picture and furthermore added an explanation to the caption

T) CAPTION: Conceptual model describing the different boundary layer processes in the valley on 16 August 2014. (a) The temporal evolution of the different layers and wind systems is given . The coloured areas refer to the distinct layers detected. Hatched areas describe distinct processes described in the text, and red arrows in the legend show the main wind direction in the respective layer. (ab) . The shows the wind systems (arrows) and the ongoing processes (shaded areas) leading to the next stage for morning, noon, afternoon, and evening, are shown in (b).

*C15 - Typo: p. 2, line 4: depend.*

A) corrected

*C16 - Typo: p.2, line 9: "and MAP" in place of "or MAP".*

A) corrected

*C17 - Typo: p. 5, line 15: drop comma after "hereby".*
A) corrected

**References**

Whiteman, C. (2000). *Mountain Meteorology: fundamentals and applications*. Oxford Univ. Press, New York.

Zardi, D. and Whiteman, C. D. (2013). Diurnal Mountain Wind Systems. In Chow, K. F., De Wekker, F. S., and Snyder, J. B., editors, *Mountain Weather Research and Forecasting: Recent Progress and Current Challenges*, pages 35–119. Springer Netherlands, Dordrecht.

---

## Author Comment (AC3) · 7 Nov 2018

Thank you very much for your comment and pointing us to a discrepancy in the manuscript. It is correct that we converted all times from UTC to Israeli Standard Time (LT = UTC + 2 h) this was done for the Radiosondes in Figure 8 and also for the model data mentioned in page 10. Only when describing the Radisonde launches on Page 4, lines 20-21 I accidentally used the Israeli Summer Time (LST= UTC+3h). I corrected the corresponding line, so the manuscript is consistent throughout. To summarise, all data (measurements + model) were converted to Israeli standard time.